# Neural signal analysis with memristor arrays towards high-efficiency brain–machine interfaces

Zhengwu Liu [1], Jianshi Tang [1,2✉], Bin Gao [1,2], Peng Yao[1], Xinyi Li[1], Dingkun Liu[3], Ying Zhou[1], He Qian[1,2], Bo Hong [3✉] & Huaqiang Wu [1,2✉]

Brain-machine interfaces are promising tools to restore lost motor functions and probe brain functional mechanisms. As the number of recording electrodes has been exponentially rising, the signal processing capability of brain–machine interfaces is falling behind. One of the key bottlenecks is that they adopt conventional von Neumann architecture with digital computation that is fundamentally different from the working principle of human brain. In this work, we present a memristor-based neural signal analysis system, where the bio-plausible characteristics of memristors are utilized to analyze signals in the analog domain with high efficiency. As a proof-of-concept demonstration, memristor arrays are used to implement the filtering and identification of epilepsy-related neural signals, achieving a high accuracy of 93.46%. Remarkably, our memristor-based system shows nearly 400× improvements in the power efficiency compared to state-of-the-art complementary metal-oxide-semiconductor systems. This work demonstrates the feasibility of using memristors for high-performance neural signal analysis in next-generation brain–machine interfaces.

[1] Institute of Microelectronics, Beijing Innovation Center for Future Chips (ICFC), Tsinghua University, Beijing 100084, China. [2] Beijing National Research Center for Information Science and Technology (BNRist), Tsinghua University, Beijing 100084, China. [3] Department of Biomedical Engineering, School of Medicine, Tsinghua University, Beijing 100084, China. ✉email: jtang@tsinghua.edu.cn; hongbo@tsinghua.edu.cn; wuhq@tsinghua.edu.cn

Brain–machine interfaces (BMIs) construct new paths between the brain and the target effectors, holding promise for the restoration of speech or motor function and the treatment of many brain disorders like epilepsy and Parkinson's diseases[1–7]. Typical BMIs record electrical signals from brain activities, using neural probes with hundreds or more recording sites[8,9], and translate them to control commands for effectors. The signal processing modules in most existing BMIs are based on silicon-based complementary metal-oxide-semiconductor (CMOS) technology and adopt the conventional von Neumann architecture where memory and data computing units are physically separated. They usually first convert analog neural signals to digital signals and then compress[10,11] and process them in the digital domain[9,12] using various application-specific integrated circuits (ASICs). Based on this approach, various interesting demonstrations have been made[5,6,10,11,13–15]. However, the design of such systems is still facing many challenges, such as power budget, delay and scalability, especially in order to catch up with the exponentially increasing number of recording sites in state-of-the-art neural probes[12,13,16–18]. Moreover, this conventional approach is fundamentally different from how brain processes information that is in analog and continuous fashion. The conversion and compression might not only cause substantial power consumption and delay[19,20] but also result in information loss, which would decrease processing accuracy. Alternatively, bioinspired and biomimetic design strategies have been recently used for invasive neural probes towards structurally and functionally stable interfaces[21–24]. These inspire us to leverage new devices that are analogous to the brain to process large-volume analog neural signals for future BMIs.

For this purpose, memristors could provide an appealing platform because they rely on field-driven ion movements to modulate their conductance, which emulate the biological behaviors of synapses and neurons in the brain[25–32]. First, as a non-volatile memory, memristors have been demonstrated to be highly efficient for in-memory computing[25,33], which is similar to the in situ information processing in the brain. Second, memristors could directly process analog signals[34,35], and parallel computing is also feasible in the form of cross-point arrays[36–38]. Last, but equally important, memristors have been shown to be fast and highly scalable down to a few nanometers[39], which could enable high-throughput processing of large-volume neural signals. Therefore, bio-plausible memristors could be a natural bridge between the brain and external circuits for future BMIs.

In this work, we propose a memristor-based neural signal analysis system for next-generation BMIs. As a proof-of-concept demonstration, we use memristor arrays to implement both long-tap finite impulse response (FIR) filter bank (as a signal preprocessor) and perceptron neural network (as a decoder) in one model system to demonstrate filtering and identifying epilepsy-related brain activities. Owing to the excellent I–V linearity and analog switching behaviors of our memristors, the system achieves a high accuracy over 93.46% while showing more than two orders of magnitude advantage in power efficiency compared to state-of-the-art CMOS systems.

## Results

### Design of memristor-based neural signal analysis system.

Figure 1 and Supplementary Fig. 1 schematically illustrate the memristor array-based neural signal analysis system and the design of a complete BMI by integrating it with neural probes. The memristor array has the central role in such a BMI as it translates the neural signals into control commands for the external effectors, such as a prosthesis or a mouse. The array of memristors with analog switching behaviors, where the device

conductance can be continuously tuned, could carry out parallel analog computing via physical laws. It thus provides a useful hardware platform to run various signal analysis algorithms while taking advantages of its high parallelism and efficiency in analog computing. For instance, memristor arrays can implement not only signal preprocessing, such as time-domain filtering and time-frequency spectrum analysis, but also decoding, which can be considered as a classification or regression task. As the preprocessor and decoder are typically the most critical and computation–extensive components in BMIs, their high-efficiency implementation would help enhance the performance and scalability of BMIs with multiple recording sites.

As a proof-of-concept demonstration of the proposed system, we construct memristor-based FIR filter bank as preprocessor and memristor-based single-layer perceptron neural network as the decoder to fulfill a typical BMI task, that is, identifying epilepsy-related brain states from recorded neural signals (Fig. 2a and Supplementary Figs. 2, 3).

FIR filter bank is an important tool for neural signal processing, and has been widely used in various biomedical applications, such as motor imagery-based BMIs[40], epilepsy detection[20], and speech synthesis[3]. The design of FIR filters is one of the bottlenecks in conventional neural signal processing ASICs because of high power and delay[20,41]. In our system, the FIR filter bank can be implemented by memristor array, which has the advantage of parallel analog computing so that the results of multiple FIR filters can be computed at one time, significantly reducing the computation power and delay. There have been proposals of designing FIR filters using a memristor crossbar structure; however most of them remain on the simulation level[42,43], and the experimental demonstration so far has been limited to a single 6-tap FIR filter using only six memristors[44]. In this work, we experimentally implement a long-tap FIR filter bank on a memristor array, which is more useful in practical applications. As the filter bank coefficients are stored in the memristor array, the output currents under input voltages represent the filter results. The basic principles of FIR filter bank implementation are described in the "Methods" section.

In the selection of neural signals for analysis, we choose local field potential (LFP) because they are found to be very useful in biomedical application, such as disease diagnosis[15,45,46] and BMIs[41,47,48]. Besides, the relatively low sample frequency makes them suitable for real-time processing with low-power electronics in BMIs[5,47]. For the system demonstration, we use the neural signals from the widely used Bonn Epilepsy Dataset (see "Methods" for more information), which are LFP signals recorded in real-world setting, for identifying epilepsy-related brain states (normal, interictal, and ictal) using our memristor-based system. The normal state indicates the subject is normal and has no epileptic neural activity. Both interictal and ictal states could be observed from epilepsy patients. The former means the patient is during the interval between epileptic seizures while the latter shows an epileptic seizure is happening inside the brain. It should be noted that, neural probes, which are commercially available, are not experimentally integrated here to complete the BMI as they are not the focus of this study. Besides, since algorithms like FIR filter are generic for various signal processing, we expect our memristor-based system to be able to seamlessly work with many different types of neural probes.

Frequency-related information in neural signals could help distinguish different brain states. There is evidence that the brain dynamics are related to neural signals in several specific frequency bands including delta ($\delta$) band (0.5~4 Hz), theta ($\theta$) band (4~8 Hz), alpha ($\alpha$) band (8~12 Hz), beta ($\beta$) band (12~30 Hz), and so on[41,49,50]. Prior works[41,51] have shown that epilepsy-related brain activities can be reflected in $\delta$, $\theta$, $\alpha$, and $\beta$ bands. So

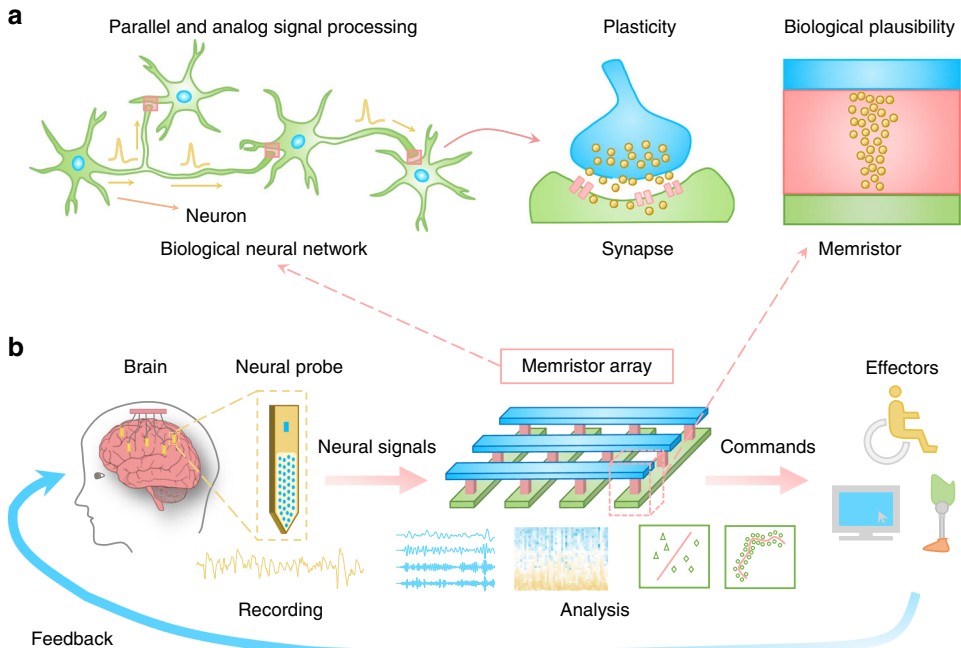

**Fig. 1 Memristor-based neural signal analysis system for brain–machine interfaces (BMIs). a** Motivations of using memristors for neural signal analysis. Memristor is analogous to biological synapse as they both show synaptic plasticity (change of its current state) via ion movements, and it has the ability to process analog signal directly as the neurons and synapses do. The memristor array further enables parallel processing of signals, which is one of the key features of the brain. **b** Conceptual diagram of a future BMI that integrates memristor-based neural signal analysis system. Neural activities recorded by the neural probes are analyzed through the memristor array to obtain commands for effectors in various BMI applications.

an FIR filter bank with four band-bass filters is designed and then implemented using memristors to generate the waves in corresponding frequency bands (Fig. 2a). Figure 2b shows how a neural signal is filtered in the memristor array. The coefficients of the designed filters are first mapped onto the memristor array as the device conductance values. We set the filter order as 120. As a result, 242 memristors are utilized to represent one filter with 121 coefficients, and 968 memristors are involved for the entire filter bank (see "Methods" and Supplementary Fig. 4 for more details of the filter design). A clip of analog voltage signal that contains the information of the brain state (i.e., normal, interictal, or ictal) is then applied to the memristor array. The sums of output currents are the filtered results from the filter bank at each time step. In this manner, the memristor array filters the input neural signals into the four frequency bands ($\delta$, $\theta$, $\alpha$, and $\beta$), whose waveforms reflect the corresponding brain state.

After that, several biomarkers, such as the waveform amplitude and energy at each frequency band, are extracted as important features (see "Methods") to be fed into a single-layer perceptron neural network to identify the epilepsy-related brain state[49,50]. The implementation of this neural network can be realized in another memristor array. Here, the weights of the neural network are trained offline and then mapped onto the memristor array (see "Methods" for more details of the neural network design). The inference process of the neural network is illustrated in Supplementary Fig. 3. The input voltages representing the extracted biomarkers are applied to the memristor array to obtain the output current vector, where the amplitudes of each element are used to identify the corresponding brain state.

**Device characterizations of analog memristors**. To realize the filtering and identification of analog neural signals with high accuracy, memristors with good analog switching behaviors and current-voltage ($I$–$V$) linearity are required. Here we use a 1k-cell array of TiN/HfO$_x$/TaO$_y$/TiN memristors in one-transistor-one-

resistor (1T1R) cell structure to implement the neural signal analysis system (Supplementary Fig. 5a–c). Our memristor device shows excellent bidirectional analog switching behavior (Fig. 2c), which enables the device conductance to be tuned continuously in both SET and RESET processes. It allows us to map the filter coefficients accurately and reconfigure the upper and lower cutoff frequencies of the filter conveniently. To demonstrate the excellent programmability of our memristor array, the letter "BMI" is written by mapping 2006 devices to different conductance states (Supplementary Fig. 5d).

In addition, good $I$–$V$ linearity in different conductance states is achieved (Fig. 2d). Importantly, the linear $I$–$V$ characteristics ensure that the memristors exhibit the same device conductance under different read voltages; otherwise, there would be errors in the processing results. It also allows us to directly use analog voltages as inputs while avoiding the conversion of neural signals to reduce the power consumption and computation delay. It should be noted that this is distinct from the conventional method of using the number of voltage pulses as digital inputs to encode the information for computation, which has been done in many previous works[19,36,38].

**Filtering epilepsy-related neural signals**. Following the above-described procedures, the FIR filter bank is first implemented in the memristor array. Figure 3a shows the conductance map (in the range of 2~20 μS) of the array for the implementation of the filter bank. Figure 3b compares the measured and target differential conductance values, showing good consistency between these two and hence excellent mapping results. Figure 3d, e and Supplementary Fig. 6 display the filtered results of typical normal, interictal and ictal neural signals in Fig. 3c that are randomly selected from the dataset. The software and experimental results match well with each other, and the difference between these two, which are mainly owing to the non-ideal device characteristics, are analyzed in Fig. 3f, g. The average error for the four filters is

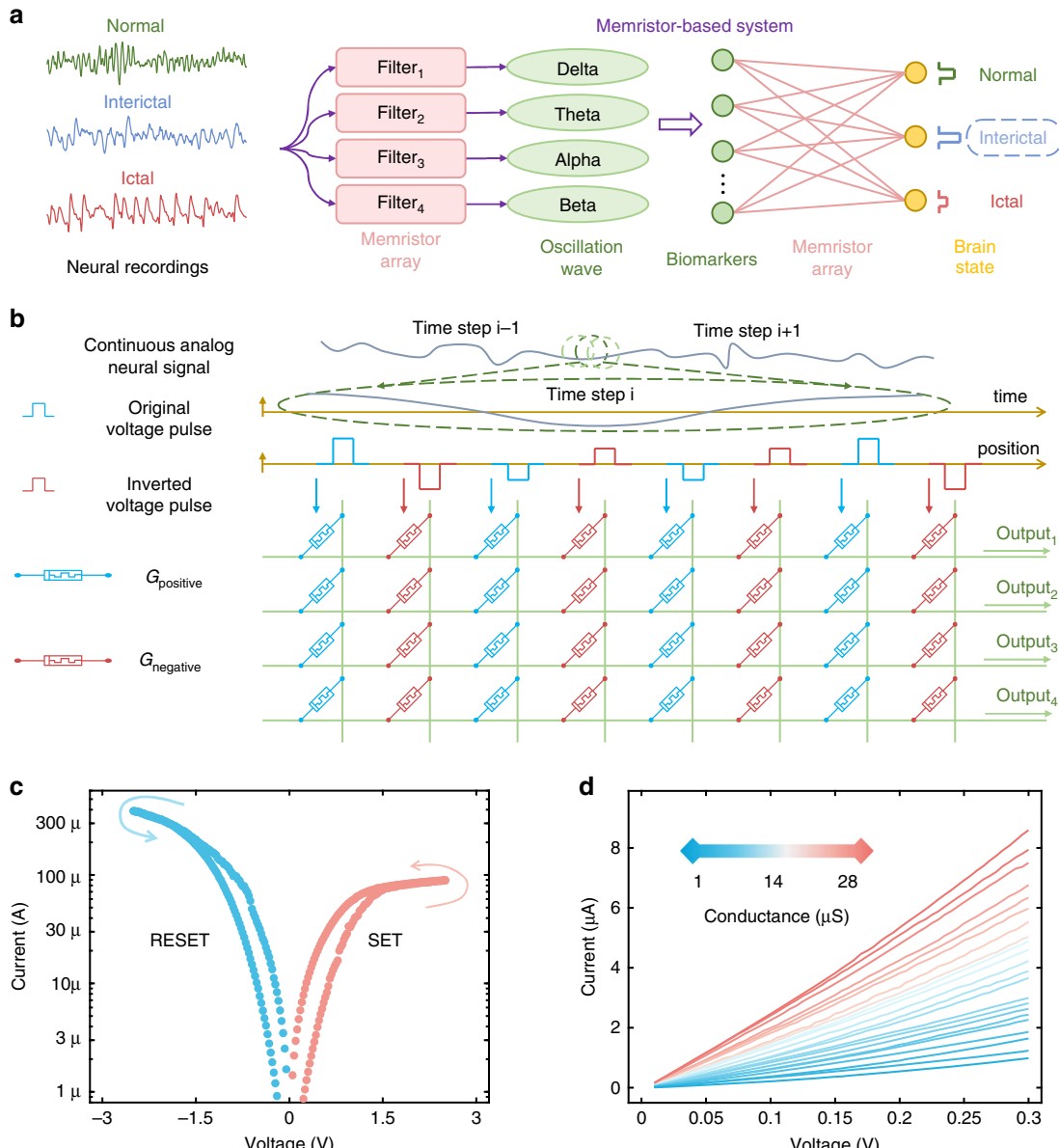

**Fig. 2 The implementation of memristor-based neural signal analysis system. a** Schematic of the memristor-based system to identify epilepsy-related brain state. Normal, interictal, and ictal neural activities are filtered by the memristor array to obtain oscillation waves in $\delta$, $\theta$, $\alpha$, and $\beta$ frequency bands. Biomarkers extracted from the filtered waves are used to identify the brain state through a single-layer neural network, which is implemented in another memristor array. **b** Implementation of the filter bank for neural activities in a memristor array. Continuous analog neural signal is conditioned and sampled as voltage pulses, which are applied to the input columns of the memristor array. Here $G_{positive}$ and $G_{negative}$ represent the memristor conductances for positive and negative weights, respectively. **c** A typical d.c. $I$–$V$ curve for a single memristor, showing excellent analog switching behaviors in both SET and RESET processes. **d** Current-voltage ($I$–$V$) characteristics read at different conductance states, showing good $I$–$V$ linearity. Source data are provided as a Source Data file.

$-0.1\ \mu V$ and the standard deviation is $1.3\ \mu V$. As we can see from these figures, the average error and standard deviation are small enough compared with the amplitude of input signals for the filtered results of all the four filters. For example, the standard deviation is <1.7% of the peak-to-valley value of the normal signal, whereas the average error is just ~1.0%. Therefore, the filtered results retain sufficient information of the input neural signals, which then can be used to faithfully identify the brain state as to be demonstrated later.

**Identifying brain states from filtered signals**. To validate the idea that our filtered results has retained sufficient information

for identifying brain state, we further construct a single-layer perceptron neural network with 21-input neurons and the output neurons in another memristor array. To compare the performance of software-calculated and memristor-based filter banks, feature vectors are extracted from both software-calculated and memristor array-filtered waveforms. Each feature vector of biomarkers corresponding to a clip of the input neural signal (see "Methods") is fed as input voltage pulses for inference. All the biomarkers extracted from the software-calculated results constitute the dataset **S**, whereas all the biomarkers extracted from the memristor array-filtered waveforms constitute the dataset **M**. Each extracted biomarker dataset contains 1800 samples, which

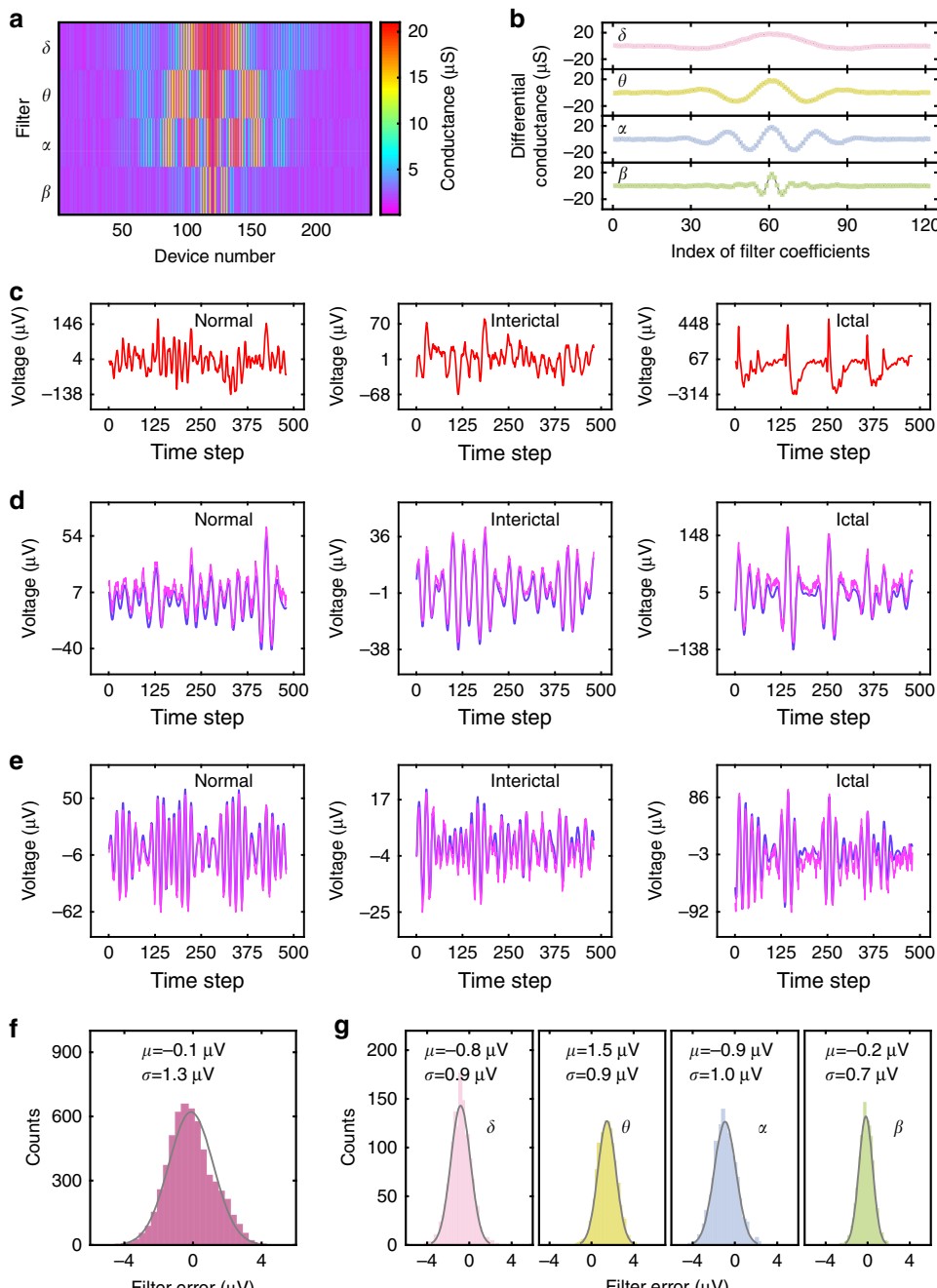

**Fig. 3 Results of the memristor array-based neural signal filter bank. a** Conductance map for the filter bank, and every column has 242 memristor devices for each 120-order filter. The filters δ, θ, α, and β are used to obtain the corresponding oscillatory waveforms. **b** Comparison of the mapped and target differential conductances for the filters δ, θ, α, and β in **a**. Software: gray lines. Measured: colored squares. **c** Typical epilepsy-related raw neural signals are fed as the input for the filter bank. **d, e** The filtered results of the filters θ and α, respectively, are shown as examples. The blue line represents software-calculated results and the purple line represents experimental results. The complete results for all the 4 filters are shown in Supplementary Fig. 6. **f, g** Error analysis between the software-calculated and the memristor array-filtered results. From left to right, the total error distribution **f** for all four filters and individual error distribution **g** for the filters δ, θ, α, and β are plotted, respectively. Here μ and σ are the mean and standard deviation, respectively. Source data are provided as a Source Data file.

are labeled as "normal", "interictal", or "ictal" accordingly. Two neural networks with the same structure are trained and tested on both datasets.

Figure 4a shows the input feature vectors of 540 testing samples. The memristor array-based neural network uses 126 devices to implement 63 synapses with differential weights, and their conductance map after training on dataset **M** is shown in Fig. 4b. Figure 4c displays the output values for the 540 testing

samples. The identification accuracies of the neural networks with software-trained weights using dataset **S** (S.S.), software-trained weights using dataset **M** (M.S.) and experimentally mapped weights after training on dataset **M** (M.M.) are compared in Fig. 4d. As we can see, the M.S. simulation has achieved nearly the same accuracy as the S.S. simulation (95.78% ± 0.27% versus 96.41% ± 0.39%, both of which are averaged from 10 trials). This result affirms that the memristor array-based filter bank retains

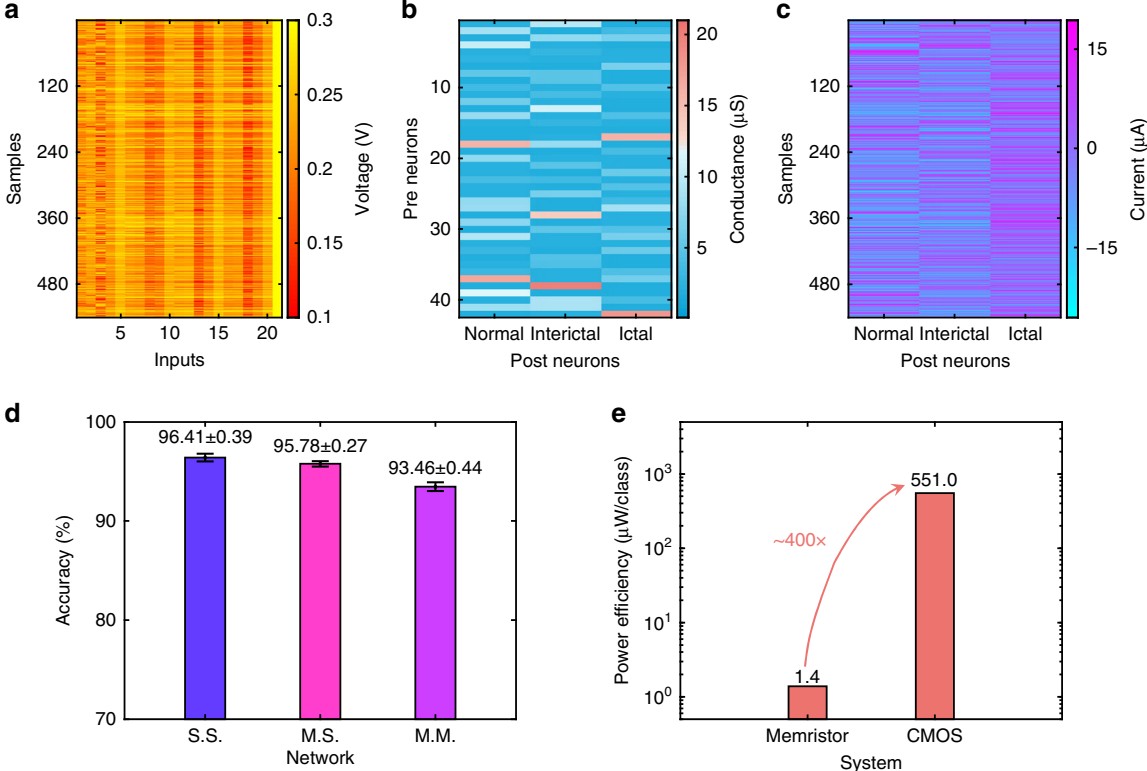

**Fig. 4 Results of the memristor array-based neural network for neural signal identification. a** Input voltages transformed from the extracted biomarker vectors to be applied on the memristor array-implemented neural network. **b** Conductance map of the memristor array for the single-layer perceptron neural network. **c** Output current of the neural network. The output neuron with the largest current value shows the input signal type. **d** Overall accuracies of the software-simulated and memristor array-implemented neural networks in the identification of the brain state. Here, error bars represent the standard deviations. S.S. represents the network trained using software-calculated results. For M.S. and M.M., the networks are trained using memristor array-filtered results, and M.S. uses the software-simulated network while M.M. uses the memristor-implemented network for inference. **e** Comparison of the power efficiency for CMOS-based and memristor-based systems. Source data are provided as a Source Data file.

sufficient information for identification and it performs as well as the software implementation. In comparison, the M.M. experiment result shows a small degradation in accuracy (93.46% ± 0.44% versus 96.41% ± 0.39%), which can be attributed to the non-ideal device characteristics of the memristor array. It is expected that this issue could be alleviated by using a larger neural network[52] or new training strategies[53].

**Performance evaluation and comparison**. Similar to state-of-the-art CMOS designs, the effect of noise is considered in our memristor-based system. In the hardware implementation of our FIR filter bank and perceptron neural network using our memristor arrays, both the filter coefficients and synaptic weights are represented by the device conductance of memristors. Therefore, the noise effects, such as read noise, conductance variations, and fluctuations, have already been incorporated in the filtered and classification results. For example, Supplementary Fig. 7 shows the memristor conductance distribution of eight typical conductance levels during mapping and read operations. In addition, the neural signals used in this work are adopted from the widely used Bonn Epilepsy Dataset, and they already captured background noises during real-world recording, including instrument noise and neural signal inherent noise. In the presence of those non-ideal factors, the achieved high accuracy in identifying epilepsy-related brain states in this work strongly indicates that our memristor-based system is robust to various noises in the hardware implementation.

Besides the achieved high accuracy, memristor-based analog computing system also provides an appealing platform to design

low-power and high-efficiency neural signal analysis system for BMIs. To compare the performance of memristor-based system with state-of-the-art CMOS-based ASICs, we estimate the power efficiency of both systems as shown in Fig. 4e. The memristor array-based system achieves a superior power efficiency of 1.4 μW/class, where most of the power is consumed by the filter bank as the size of the perceptron network for identification is relatively small. In comparison, a typical CMOS system is estimated to have a power efficiency of 551.0 μW/class (see "Methods" for details). As a result, our memristor-based system shows a ~400× advantage in the power efficiency compared with state-of-the-art CMOS systems.

Such advantage mainly comes from the fact that analog neural signals can be directly processed on memristor arrays without the need of neural signal conversion and compression that inevitably consumes significant power and energy. At the same time, the in-memory computing capability of memristors also minimizes the energy and time needed for fetching the filter coefficients or weights in conventional von Neumann architecture. These unique merits make memristor arrays an appealing candidate for high-throughput analog neural signals analysis in future fully implanted BMIs.

## Discussion
In summary, we have proposed a memristor-based neural signal analysis system with high efficiency for future BMIs. Memristor arrays are used to implement the filter bank and neural network to demonstrate the filtering and identification of epilepsy-related neural signals and brain states. The memristor-based system has

achieved a high accuracy of ~93.46% while achieving a nearly 400× advantage in power efficiency compared to state-of-the-art CMOS systems. Such computation advantages could be further enhanced by future device optimization to address the non-ideal characteristics. Our work experimentally demonstrates the potential of memristor-based systems for low-power and high-energy-efficiency in situ analysis towards next-generation BMIs that could have millions of neural recording sites. Future work is required to prototype a fully implanted BMI by monolithically integrating multi-functional memristor-based signal analysis modules with state-of-the-art neural probes.

## Methods

**Epilepsy-related signals**. In the task of filtering and identifying epilepsy-related signals, all the neural signals of volunteers or patients in normal, interictal, and ictal periods are from University Hospital of Bonn[46]. There are 100 neural signal clips in total for each class and all the clips from the above three classes are used. Each clip of raw neural signals has 4096 samples and is divided into six segments, where each segment contains 600 samples and the rest 496 samples are discarded. In this demonstration, as Supplementary Fig. 8 illustrates, the signals represented by the values with digitalized levels in the dataset are transformed to voltage pulses with analog amplitudes to be applied on the memristor array.

**Memristor array fabrication**. One 1T1R cell consists of an NMOS transistor whose drain is connected to a resistive switching memory to serve as the selector. For the memristor array used in this work, the transistor array is fabricated in a commercial foundry using 0.13 μm standard CMOS process, and the memristor has a material stack of TiN/HfO$_x$/TaO$_x$/TiN. Details about the device fabrication can be found in our previous works[36,54].

**Memristor array-based FIR filter bank**. In general, a filter bank consisting of multiple FIR filters can be mathematically expressed as[55]:

$$y^m(n) = \sum_{k=0}^{K} x(n-k)h^m(k), (m=1,2,...,M) \qquad (1)$$

where $x$ represents the input neural signal. $k$ and $K$ are the filter coefficient index and the filter order, respectively. $n$ is the index of time step. $m$ and $M$ are the sequence number and the total number of filters respectively. $h^m$ represents the impulse response of the $m$th filter, whose pattern determines the property of the $m$th filter. $y^m$ represents the filtered signal of the $m$th filter.

To implement the filter bank in memristor array, Eq. (1) can be re-written as follows:

$$y_n = x_n \mathbf{H} \qquad (2)$$

where $x_n$ and $y_n$ are the input and output signal row vectors in the $n$th time step, respectively. The matrix $\mathbf{H}$ represents all the filter coefficients for the filter bank. Elements in the $m$th column of $\mathbf{H}$ are the coefficients of the $m$th filter, and they can be represented by the device conductance in a memristor array for hardware implementation. To implement a filter bank whose filter coefficients may have both positive and negative values, we use two memristors as a differential pair to represent one coefficient. In this manner, the implementation of the filter bank in a memristor array can be expressed as:

$$I^m(n) = \sum_{k=0}^{K} \left\{ V(n-k)\left(G_+^m(k) - G_-^m(k)\right) \right\} = \sum_{k=0}^{K} \qquad (3)$$
$$\left\{ V(n-k)G_+^m(k) + [-V(n-k)]G_-^m(k) \right\}, (m=1,2,\dots,M)$$

where $V$ is the input voltage vector, and $\left(G_+^m(k) - G_-^m(k)\right)$ represents the mapped element at the cross-point of the $k$th row and $m$th column of the filter coefficients matrix $\mathbf{H}$. $I^m$ is the output current vector of the $m$th filter.

**Memristor-based filter design**. All the filters are first designed using MATLAB (version: 2018b). The filter order is chosen by comparing different waveforms of the filtered signals with the same parameter settings, i.e., the same lower and upper cutoff frequencies, but with different orders from 40 to 200 (see Supplementary Fig. 4).

**Memristor-based neural network**. We use a 21 × 3 single-layer perceptron neural network to identify the signal type from the filtered results. For each signal in the dataset, to extract the 20 biomarkers/features from the output currents of the filter banks, first, the output currents are amplified and offset to obtain the filtered voltage waveforms. Then we calculate the maximum value, minimum value, mean value, sum of absolute value and sum of energy for the voltage waveforms to obtain the 20 biomarkers (five features for each of four waveforms). These computations are currently done by software in this work for simplicity but in principle they could also be implemented by memristor-based electronics[56,57]. Furthermore, we normalize the extracted biomarkers to the range of 0.1–0.3 V by software using a

linear transformation (e.g., $y = a \times x + b$, $x$: input; $y$: output; $a$, $b$: amplification factor and offset), and then use them as the input voltages for the memristor-based neural network. The dataset contains 1800 samples, i.e., 600 samples for each signal type. In all, 30% of the total dataset is used as the testing set and the remaining 70% forms the training set. During the training process, we use the sigmoid activation function and the backpropagation algorithm is used to train the weights in neural network.

**Power efficiency estimation**. The power efficiency is related to the sampling frequency of the neural signals and also the number of recording channels. Here, a one-channel 0.1 s signal clip, which is sampled at 10 kHz is as the standard signal for power estimation. Then for 1 s duration, there are 10 standard signal clips to be processed. The power of a read operation for a memristor is calculated as $(0.2V)^2 \times 20\,μs = 0.8\,μW$. When the filter bank that includes four 120-order (121-tap) filters is implemented using differential pairs, $(0.8\,μW \times 121 \times 2 + 0.50\,mW) \times 50\,ns \times 4 = 138.7$ pJ/sample is calculated as the power for each sample point. Here, 0.50 mW is the estimated power of trans-impedance amplifier (TIA) to convert the output current of memristors (in the typical range from $-40\,μA$ to $100\,μA$) into voltage. Then the power efficiency for filtering is estimated as 138.7 pJ/sample × (1000 samples)/(1 s/10) = 1.39 μW/class. For the signal-layer perceptron neural network with 21 ×3 = 63 weights, the power efficiency is calculated as $(0.8\,μW \times 21 \times 2 + 0.10\,mW) \times 3 \times 50\,ns/(1\,s/10) = 0.20$ nW/class, where 0.10 mW is the estimated TIA power for each output neuron (current in the typical range between $-5\,μA$ and $5\,μA$). A typical advanced FIR filter design for neural signal preprocessing based on CMOS technology (under 130 nm technology node) is described in ref. [20]. This work includes a total effective number of 64 16-tap FIR filters for neural signal filtering and these filters consume 0.53 mW when processing neural signals sampled at a frequency of 7.1 kHz. If they are directly scaled up to a filter bank with four 121-tap filters working at 10 kHz, then the power efficiency is estimated as 0.53 mW/(64 × 16) × (4 × 121)/(7.1 kS/s) × 10 kS/s = 352.0 μW / class. For a typical decoder designed with CMOS technology, described in ref. [14], 273 μJ is consumed by a support vector machine classifier with a (18 × 3 × 8)-inputs feature vector for every 2 s signal epoch. So, if it is scaled to a classifier with 21-input dimension and works at the classification rate of 10 Hz, the power efficiency is estimated as (273 μJ/(2 s))/(18 × 3 × 8 inputs) × 21 inputs/(2 class) × 3 class/(0.5 Hz)×10 Hz = 199.0 μW/class. From the above estimations, the overall power efficiency is ~1.4 μW/class for the memristor-based system and 551.0 μW/class for the CMOS-based system.

## Data availability

The source data for Fig. 2c–d, 3, and 4 are provided in separate Source Data files. Other data that support the findings of this study are available from the corresponding author upon reasonable request. Source data are provided with this paper.

## Code availability

The codes that support the findings of this study are available from the corresponding author upon reasonable request.

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

## Acknowledgements

This work was supported in part by China key research and development program (2019YFB2205403, 2017YFA0205904), and Natural Science Foundation of China (61851404, 61974081, 91964104).

## Author contributions

Z.L., J.T., B.G., and H.W. conceived and designed the experiments. Z.L. and P.Y. performed the experiments. Z.L. and Y.Z. contributed to the simulation. Z.L., J.T., B.G., X.L., D.L., B.H., and H.W. contributed to the paper writing. All authors discussed and reviewed the manuscript. H.Q., H.W., and J.T. supervised the project.

## Competing interests

The authors declare no competing interests.
