## [Peer Review File · Nature Communications]

Reviewers' comments:

Reviewer #1 (Remarks to the Author):

The submitted article proposes an array of a memristor-based finite impulse response (FIR) filters for neural signal identification and analysis. In the proposed system, neural probes are used to record neural signals and the array filter is used to analyze it. The proposed system is crucial for the brain-machine interface system (BMI).

The reviewer comments are as follows:

- The novelty of the paper is still missing. This is not the first paper about using memristors in pattern classification or neuromorphic computing.
- The comparison with the state of the art CMOS is not a valid comparison since it is comparing with a complete CMOS design with high noise immunity. The proposed system does not provide how much the system is immune to different types of noise.
- The paper claims proposing a novel BMI system combining neural probes with memristor arrays to integrate neural signal detecting and in situ analysis, while the neural probes are not integrated within the proposed design. There was no mention of the fabrication or integration of the neural probes with the memristor array at the method section. Does the paper propose an FIR filter design using a memristor array for neural signal analysis? This is not the first time an FIR filter is proposed using a memristor crossbar structure.
- The reviewer recommends the paper to be restructured since the flow of the paper is inconsistent with the recommended flow to ease the read and understanding of the paper. The following comments might help.
 - #- The classification of neural signals is not well defined. The transition from each paragraph to the other and section to the other is not harmonic and consistent.
 - #- The contribution of the paper is not well defined. The paper is too ambitious about future applications than to describe the actual work itself.
 - #- The Results section contains the theory behind the FIR filter which may be defined earlier in a separate section.

Reviewer #2 (Remarks to the Author):

This manuscript reports the implementation of two important components, filter banks and an artificial neural network, of an overall BMI system using memristor-based electronics. As a demonstration, such electronics is used to process the analog ECoG signals for classification of epilepsy states. The acclaimed advantages of this approach include higher efficiency and lower power consumption comparing to conventional CMOS electronic implementations.

The overall concept is cool and appealing, while the demonstration shows certain advantages.

Some concerns are listed below:

- (1) The idea presented in the manuscript is a BMI system implementable by the memristor-based electronics, but only two components of such a system is implemented and demonstrated in use. No BMI is involved at all, which diminishes the enthusiasm conveyed in the introduction.
- (2) The two claimed advantages, higher efficiency and lower power consumption, of the memristor-based electronics are not directly supported by the experimental data. Ideally, the control should be the conventional CMOS circuit that implement the same functions. It is hard to access how good the accuracy of 93.46% is, as the dataset may lead to similar level of classification accuracy independent from the specific implementation platforms. The analysis on the power consumption comparing to an equivalent circuit implemented using CMOS only provides indirect evidence.
- (3) The analog processing capability of the memristor-based electronics is used, but the Bonn Epilepsy

dataset should be digital signals. How were these signals converted to analog in the first place? How were the 20 biomarkers/features extracted from the output currents of the filter banks? Could this extraction be implemented by the memristor-based electronics? If not, the concept of a memristor-based BMI is diminished. How were the input voltages representing the extracted biomarkers to the artificial neural network obtained from the features of the output currents of the filter banks?

(4) Equations 1-3 are for discrete-time, continuous-valued signals; particularly, Equation 1 is the discrete-time convolution sum. How were such discrete-time "analog" input signals ($x[n]$) obtained from the digital Bonn Epilepsy dataset.

(5) The first sentence of the Conclusion reads "In summary, we have proposed a novel BMI system...", however, such a BMI system implemented entirely or primarily using the memristor-based electronics is not demonstrated. Furthermore, no integration of signal-recording neural probes with signal-analysis memristor arrays is demonstrated.

Response Letter to Reviewers' Comments

We sincerely appreciate the valuable time the reviewers have spent reviewing our manuscript and providing insightful comments and suggestions to help further improve the quality of our work. We believe we have addressed all of the reviewers' comments and now the paper is more rigorous in content and clearer in presentation. Our point-by-point responses to the reviewers' comments are as follows.

Reviewer #1

The submitted article proposes an array of a memristor-based finite impulse response (FIR) filters for neural signal identification and analysis. In the proposed system, neural probes are used to record neural signals and the array filter is used to analyze it. The proposed system is crucial for the brain-machine interface system (BMI).

Response:

Thank you for recognizing the significance of our work and the importance of the proposed memristor-based system for BMI applications. Our detailed responses to your technical comments are provided below.

Comment #1:

The novelty of the paper is still missing. This is not the first paper about using memristors in pattern classification or neuromorphic computing.

Response:

Thank you for your comment. As we clearly discussed in the Introduction part, we agree with the reviewer that memristors have been widely used for pattern classification and neuromorphic computing in literature (e.g., Li et al., *Nature Communications*, 2018; Ambrogio et al., *Nature*, 2018; Sun et al., *Proceedings of the National Academy of Sciences*, 2019; Yao et al., *Nature*, 2020). However, there have been no reports of **using two memristor arrays to implement both long-tap FIR filter bank (as a signal pre-processor) and perceptron neural network (as a decoder) in one model system for neural signal analysis and classification**, which is the key novelty of this work. This is a more complete neural signal analysis system than existing ones with either signal processing only or pattern classification only. As a proof-of-concept demonstration, for the first time, **we demonstrated the identification of epilepsy-related brain states in our memristor arrays using neural signals** from the widely used Bonn Epilepsy Dataset. We have optimized the I-V linearity and analog switching behaviors of our memristors over a large array to **achieve the software-equivalent accuracy and high power efficiency in this work**. We believe our work represents a significant milestone towards developing viable bio-inspired memristor-based neural signal analysis modules for future fully implanted BMI systems.

To avoid any confusion, we have further clarified the novelty of this work in the revised manuscript as follows:

On page 1, line 21: "...a memristor-based neural signal analysis system, ..."; On page 1, line 23: "As a proof-of-concept demonstration, ..."; On page 3, lines 65 - 70: "we propose a memristor-based neural signal analysis system for next-generation BMIs. As a proof-of-concept demonstration, we use memristor arrays to implement both long-tap finite impulse response (FIR) filter bank (as a signal pre-processor) and perceptron neural network (as a decoder) in one model system to demonstrate filtering and identifying epilepsy-related brain activities. Owing to the excellent I-V linearity and analog switching behaviors of our memristors, the system ..."; On page 3, lines 76-78: "... the memristor array-based neural signal analysis system and the design of a complete BMI by integrating it with neural probes. ... in such a BMI as it ..."; On page 4, lines 86 - 89: "Since the preprocessor and decoder are typically the most critical and computation-extensive components in BMIs, their high-efficiency implementation would help enhance the performance and scalability of BMIs with multiple recoding sites."; On page 10, lines 254 – 255: "..., we have proposed a memristor-based neural signal analysis system with high efficiency for future BMIs. Memristor ...".

Comment #2

The comparison with the state of the art CMOS is not a valid comparison since it is comparing with a complete CMOS design with high noise immunity. The proposed system does not provide how much the system is immune to different types of noise.

Response:

Thank you for the comment. As a standard benchmark, similar comparisons between CMOS and memristor-based systems have been made in many prior works (e.g., Li et al., *Nature Machine Intelligence*, 2019; Wang et al., *Nature Electronics*, 2019; Yao et al., *Nature*, 2020). Similar to state-of-the-art CMOS design, noise is in fact considered in our memristor-based system. In the hardware implementation of our FIR filter bank and perceptron neural network using our memristor arrays, both the filter coefficients and synaptic weights are represented by the device conductance of memristors. Therefore, **the noise effects, such as read noise, conductance variations and fluctuations, have already been incorporated in the filtered and classification results.** For example, **Figure R1** below shows the memristor conductance distribution of four typical conductance levels during mapping and read operations.

In addition, the neural signals used in this work are adopted from the widely used Bonn Epilepsy Dataset, and **they already captured background noises during real-world recording**, including instrument noise and neural signal inherent noise. In the presence of those non-ideal factors, the achieved high accuracy in identifying epilepsy-related brain states in this work strongly indicates that **our memristor-based system is immune to various noises in the hardware implementation.**

To clarify this point, we have added **Figure R1** as **Supplementary Figure 7** in the revised **Supplementary Information** and also incorporated the above discussions in the revised manuscript:

On pages 8 - 9, lines 221 – 234: “**Similar to state-of-the-art CMOS designs, the effect of noise is considered in our memristor-based system. In the hardware implementation of our FIR filter bank and perceptron neural network using our memristor arrays, both the filter coefficients and synaptic weights are represented by the device conductance of memristors. Therefore, the noise effects, such as read noise, conductance variations and fluctuations, have already been incorporated in the filtered and classification results. For example, Supplementary Fig. 7 shows the memristor conductance distribution of eight typical conductance levels during mapping and read operations. In addition, the neural signals used in this work are adopted from the widely used Bonn Epilepsy Dataset, and they already captured background noises during real-world recording, including instrument noise and neural signal inherent noise. In the presence of those non-ideal factors, the achieved high accuracy in identifying epilepsy-related brain states in this work strongly indicates that our memristor-based system is robust to various noises in the hardware implementation.**”

Figure R1. Conductance distributions of memristors in the mapping and read operations. **a-h**, Mapped conductance distributions of eight typical levels: 2.5, 5.0, 7.5, 10.0, 12.5, 15.0, 17.5 and 20 μS . Each distribution includes the mapping results of 1000 memristors. **i-p**, Read noise distributions of these eight conductance levels. Each distribution is obtained from 100-times read results.

Comment #3.1

The paper claims proposing a novel BMI system combining neural probes with memristor arrays to integrate neural signal detecting and in situ analysis, while the neural probes are not integrated within the proposed design. There was no mention of the fabrication or integration of the neural probes with the memristor array at the method section.

Response:

Thanks for your comments. Indeed, in this paper, we propose the concept of a novel BMI system that consists of neural probes for recording and memristor arrays for signal processing. As we explained above in the response to your first comment, **the main focus and novelty of this work is to experimentally demonstrate the feasibility of using memristor arrays to implement both FIR filter bank and perceptron neural network** to realize neural signal analysis and classification with high accuracy and power efficiency. This is the most important component in our proposed BMI system. The neural probes, which have been extensively researched in the literature, are not the focus of this study and hence are not experimentally integrated yet. Instead, we use the neural signals from the widely used Bonn Epilepsy Dataset (e.g., Andrzejak et al., *Phys Rev E Stat Nonlin Soft Matter Phys*, 2001; Adeli et al., *IEEE Transactions on Biomedical Engineering*, 2007; Orhan et al., *Expert Systems with Applications*, 2011; Samiee et al., *IEEE Transactions on Biomedical Engineering*, 2015; Acharya et al., *Computers in Biology and Medicine*, 2018), which are raw neural signals recorded in real-world setting, for the proof-of-concept demonstration for identifying epilepsy-related brain states using our memristor-based system. Since algorithms like FIR filter are **generic** for various signal processing, **we expect our memristor-based system to be able to seamlessly work with many different types of neural probes**. In fact, inspired by the exciting results obtained in this work, we are currently working on integrating both neural probes and memristor arrays to build prototype BMI system for potential clinic trials, which is the very next step of the present work.

In order to clarify this point, we have added that “**For the system demonstration, we use the neural signals from the widely used Bonn Epilepsy Dataset (see **Methods** for more information), which are LFP signals recorded in real-world setting, for identifying epilepsy-related brain states (normal, interictal and ictal) using our memristor-based system. The normal state indicates the subject is normal and has no epilepsy. Both interictal and ictal states could be observed from epilepsy patients. The former means the patient is during the interval between epileptic seizures while the latter shows an epileptic seizure is happening inside the brain. It should be noted that, neural probes, which are commercially available, are not experimentally integrated here to complete the BMI as they are not the focus of this study. Besides, since algorithms like FIR filter are generic for various signal processing, we expect our memristor-based system to be able to seamlessly work with many different types of neural probes.**” on pages 5 - 6, lines 113 – 124.

Comment #3.2

Does the paper propose an FIR filter design using a memristor array for neural signal analysis? This is not the first time an FIR filter is proposed using a memristor crossbar structure.

Response:

Thanks for your comment. Yes, in this work we implemented an FIR filter bank using a memristor array for neural signal analysis. Although the idea of designing an FIR filter using a memristor crossbar structure has been proposed previously, our work experimentally implements a **long-tap FIR filter bank** for the first time, to the best of our knowledge:

- 1) In literature, there are several proposals of designing FIR filters using memristors (e.g., Nourazar et al., *Analog Integrated Circuits and Signal Processing*, 2018; Mirebrahimi et al., *Analog Integr Circ Sig Process*, 2014); however, **most of them remain on the simulation level** with no actual hardware demonstrations, as illustrated in **Figures R2a-c** below. In these simulations, ideal device model was often used without considering non-ideal device characteristics, such as I-V nonlinearity, programming and reading noises. In comparison, our experimental demonstration verified the feasibility of memristor-based FIR filter bank in the fabricated memristor array, which already includes the effect of non-ideal device characteristics in the computation processes.
- 2) The only work that presented the experimental demonstration of FIR filter is Alibart et al., *arxiv.1608.05445*, 2016, as illustrated in **Figures R2d-f** below. However, **the demonstration has been limited to a single 6-tap FIR filter using only 6 memristors**, which is not a practically useful FIR filter bank with multiple long-tap filters as demonstrated in our work (which is a more comprehensive system **using 968 memristors**). In addition, **the previous demonstration used sine wave mixed with white noises as the test signal**, which is much simpler than real-word signals like epilepsy-related human brain signals used in our work. Therefore, the implementation of FIR filter bank using memristor array for neural signal analysis in this work represents a significant advance beyond prior work.

In order to clarify the advance of our work beyond literature reports, we have cited the above reference (Refs. 42-44) and explained their difference in the “**Design of memristor-based neural signal analysis system for BMIs**” part in the revised manuscript. We have added that “**There have been proposals of designing FIR filters using a memristor crossbar structure; however most of them remain on the simulation level^{42,43}, and the experimental demonstration so far has been limited to a single 6-tap FIR filter using only 6 memristors⁴⁴. In this work, we experimentally implement a long-tap FIR filter bank on a memristor array, which is more useful in practical applications. As the filter bank coefficients are stored in the memristor array, the output currents under input voltages represent the filter results. The basic principles of FIR filter bank implementation are described in the **Methods** section.**” on page 4, lines 101-109.

Figure R2. Previous works on memristors-based FIR filters in literature. a-c, Simulation design of memristor-based FIR filters, adopted from Mirebrahimi et al., *Analog Integr Circ Sig Process*, 2014. **a**, The circuit architecture. **b**, Simulation results of a 6-tap FIR filter for filtering noisy sine waves. **c**, Simulation results of a 2-tap FIR filter for demodulating AM signals. **d-f**, Experimental demonstration of memristor-based FIR filter, adopted from Alibart et al., *arxiv.1608.05445*, 2016. **d**, The bread-board implementation architecture. **e**, Experimental results of a memristor-based 6-tap FIR filter with two different weight settings for denoising noise-full sine waves. **f**, Comparison of frequency responses for the FIR filter in the two cases.

Comment #4

The reviewer recommends the paper to be restructured since the flow of the paper is inconsistent with the recommended flow to ease the read and understanding of the paper. The following comments might help.

#- The classification of neural signals is not well defined.

The transition from each paragraph to the other and section to the other is not harmonic and consistent.

#- The contribution of the paper is not well defined. The paper is too ambitious about future applications than to describe the actual work itself.

#- The Results section contains the theory behind the FIR filter which may be defined earlier in a separate section.

Response:

Thank you for your suggestions. We have re-organized our manuscript accordingly to improve the flow and readability of this paper. Major revisions are summarized as follows:

1. To clearly define the neural signal analysis and classification task, we have added “... we use the neural signals from the widely used Bonn Epilepsy Dataset (see **Methods** for more information), which are LFP signals recorded in real-world setting, for identifying epilepsy-related brain states (normal, interictal and ictal) using our memristor-based system. The normal state indicates the subject is normal and has no epilepsy. Both interictal and ictal states could be observed from epilepsy patients. The former means the patient is during the interval between epileptic seizures while the latter shows an epileptic seizure is happening inside the brain.” on page 5, lines 114 – 120.
2. To make the connections between two sections and two paragraphs more harmonic and consistent, we have made the following revisions: we have added “**Since the preprocessor and decoder are typically the most critical and computation-extensive components in BMIs, their high-efficiency implementation would help enhance the performance and scalability of BMIs with multiple recoding sites.**” on page 4, lines 86 - 89; we have added “**As a proof-of-concept demonstration of the proposed system, we construct memristor-based FIR filter bank as preprocessor and memristor-based single-layer perceptron neural network as the decoder to fulfill a typical BMI task, that is, identifying epilepsy-related brain states from recorded neural signals (Fig. 2a and Supplementary Figs. 2, 3).**” on page 4, lines 90 - 94; we have added “**As the filter bank coefficients are stored in the memristor array, the output currents under input voltages represent the filter results. The basic principles of FIR filter bank implementation are described in the **Methods** section.**” on page 5, lines 106 - 109; we have added “**In the selection of neural signals for analysis,**” on page 5, line 110; we have added “**Frequency-related information in neural signals could help distinguish different brain states.**” on page 6, lines 125 – 126 and “...,”

the FIR filter bank is first implemented in the memristor array....” on page 7, lines 198 - 199.

3. To explicitly define the contributions of this work, we have further clarified the novelty of this work in the revised manuscript as follows: On page 1, line 21: “...a memristor-based neural signal analysis system, ...”; On page 1, line 23: “As a proof-of-concept demonstration, ...”; On page 3, lines 65 - 70: “we propose a memristor-based neural signal analysis system for next-generation BMIs. As a proof-of-concept demonstration, we use memristor arrays to implement both long-tap finite impulse response (FIR) filter bank (as a signal pre-processor) and perceptron neural network (as a decoder) in one model system to demonstrate filtering and identifying epilepsy-related brain activities. Owing to the excellent I-V linearity and analog switching behaviors of our memristors, the system ...”; On page 4, lines 86 - 89: “Since the preprocessor and decoder are typically the most critical and computation-extensive components in BMIs, their high-efficiency implementation would help enhance the performance and scalability of BMIs with multiple recoding sites.”; On page 10, lines 254 – 255: “..., we have proposed a memristor-based neural signal analysis system with high efficiency for future BMIs. Memristor ...”.
4. We have re-arranged the organizations of our manuscript. We have made the introduction of BMIs more concise in the **Introduction** section and focused on the challenges faced by the signal processing modules in most existing BMIs. We have changed the “System Design” section to the first subsection entitled “**Design of memristor-based neural signal analysis system for BMIs**” in the **Results** section. The definition of the epilepsy-related brain states identification task and the proof-of-concept system implementation have been included in this subsection. The theory and memristor array-based implementation of the FIR filter bank have become a separate subsection entitled “**Memristor array-based FIR filter bank**” in the **Methods** section. Device requirements for the implementation have been described in a new and the third subsection titled “**Device characterizations of analog memristors**” in the **Results** section.

Reviewer #2 (Remarks to the Author):

This manuscript reports the implementation of two important components, filter banks and an artificial neural network, of an overall BMI system using memristor-based electronics. As a demonstration, such electronics is used to process the analog ECoG signals for classification of epilepsy states. The acclaimed advantages of this approach include higher efficiency and lower power consumption comparing to conventional CMOS electronic implementations. The overall concept is cool and appealing, while the demonstration shows certain advantages.

Response:

We thank the reviewer for recognizing the “cool and appealing” concept of this work and the significance of the implementations of filter bank and neural network. Our detailed responses to your technical comments are provided below.

Some concerns are listed below:

Comment #1

(1) The idea presented in the manuscript is a BMI system implementable by the memristor-based electronics, but only two components of such a system is implemented and demonstrated in use. No BMI is involved at all, which diminishes the enthusiasm conveyed in the introduction.

Response:

Thank you for your comments. In this work, we propose the concept of a novel BMI system that consists of neural probes for recording and memristor arrays for signal processing. The reason why neural probes are not experimentally integrated at this stage to implement the complete BMI system is as follows:

- 1) As we discussed in the Introduction part, **the performance and scalability of existing BMI systems is largely limited by the electronics used to process neural signals** rather than the neural probes used to collect neural signals. Therefore, **the main focus and novelty of this work is to experimentally demonstrate the feasibility of using memristor arrays to implement both FIR filter bank and perceptron neural network**, which realize neural signal analysis and classification with high accuracy and power efficiency. They are also the most important components in our proposed BMI system.
- 2) To resemble real-world BMI applications, we use the neural signals from the widely used Bonn Epilepsy Dataset (e.g., Andrzejak et al., *Phys Rev E Stat Nonlin Soft Matter Phys*, 2001; Adeli et al., *IEEE Transactions on Biomedical Engineering*, 2007; Orhan et al., *Expert Systems with Applications*, 2011; Samiee et al., *IEEE Transactions on Biomedical Engineering*, 2015; Acharya et al., *Computers in Biology and Medicine*, 2018), which are **raw neural signals recorded in real-world setting**, for the proof-of-concept demonstration for identifying epilepsy-related brain states using our memristor-based system.

- 3) Since algorithms like FIR filter are **generic** for various signal processing, **we expect our memristor-based system to be able to seamlessly work with many different types of neural probes.** In fact, inspired by the exciting results obtained in this work, we are currently working on integrating both neural probes and memristor arrays to build prototype BMI system for potential clinic trails, which is the very next step of the present work.

Therefore, we believe that, as the first and key step of implementing a complete BMI system, our work represents a significant milestone towards developing viable memristor-based modules for bio-inspired neural signal analysis. Major revisions are summarized as follows:

On page 1, line 21: “...a memristor-based neural signal analysis system, ...”; On page 1, line 23: “As a proof-of-concept demonstration, ...”; On page 3, lines 65 - 70: “we propose a memristor-based neural signal analysis system for next-generation BMIs. As a proof-of-concept demonstration, we use memristor arrays to implement both long-tap finite impulse response (FIR) filter bank (as a signal pre-processor) and perceptron neural network (as a decoder) in one model system to demonstrate filtering and identifying epilepsy-related brain activities. Owing to the excellent I-V linearity and analog switching behaviors of our memristors, the system ...”; On page 4, lines 86 - 89: “Since the preprocessor and decoder are typically the most critical and computation-extensive components in BMIs, their high-efficiency implementation would help enhance the performance and scalability of BMIs with multiple recoding sites.”; On pages 5 - 6, lines 113 – 124: “For the system demonstration, we use the neural signals from the widely used Bonn Epilepsy Dataset (see **Methods** for more information), which are LFP signals recorded in real-world setting, for identifying epilepsy-related brain states (normal, interictal and ictal) using our memristor-based system. The normal state indicates the subject is normal and has no epilepsy. Both interictal and ictal states could be observed from epilepsy patients. The former means the patient is during the interval between epileptic seizures while the latter shows an epileptic seizure is happening inside the brain. It should be noted that, neural probes, which are commercially available, are not experimentally integrated here to complete the BMI as they are not the focus of this study. Besides, since algorithms like FIR filter are generic for various signal processing, we expect our memristor-based system to be able to seamlessly work with many different types of neural probes.”; On page 10, lines 254 – 255: “..., we have proposed a memristor-based neural signal analysis system with high efficiency for future BMIs. Memristor ...”.

Comment #2

(2) The two claimed advantages, higher efficiency and lower power consumption, of the memristor-based electronics are not directly supported by the experimental data. Ideally, the control should be the conventional CMOS circuit that implement the same functions. It is hard to access how good the accuracy of 93.46% is, as the dataset may lead to similar level of classification accuracy independent from the specific implementation platforms. The analysis on the power consumption comparing to an equivalent circuit implemented using CMOS only provides indirect evidence.

Response:

Thank you for the comment. As the reviewer pointed out, ideally, the same dataset and algorithms should be used to make a fair comparison between our memristor-based system and CMOS counterpart; however, **it is very difficult to find such control in practice**. There are many works in literature using the same Bonn Epilepsy dataset (Andrzejak et al., *Phys Rev E Stat Nonlin Soft Matter Phys*, 2001), but most of them studied the algorithms for epilepsy detection or classification without hardware implementation (Acharya et al., *Computers in Biology and Medicine*, 2018; Samiee et al., *IEEE Transactions on Biomedical Engineering*, 2015; Bajaj et al., *Ieee Transactions on Information Technology in Biomedicine*, 2012; Guler et al., *Journal of Neuroscience Methods*, 2005). So far, to the best of our knowledge, there are only two CMOS-based systems utilizing this dataset Crispin-Bailey et al., *IEEE Transactions on Biomedical Circuits and Systems*, 2019; Daoud et al., *IEEE Transactions on Biomedical Circuits and Systems*, 2020. However, the processing and decoding algorithms they implemented are quite different from ours in this work. Therefore, we decide to use equivalent circuits implemented with CMOS to compare the power comparison. In fact, **similar comparisons between CMOS and memristor-based systems have been made in many prior works** Li et al., *Nature Machine Intelligence*, 2019; Wang et al., *Nature Electronics*, 2019; Yao et al., *Nature*, 2020, which becomes a standard benchmark for memristor-based electronics.

For the accuracy analysis, as we have described in the **Identifying brain states from filtered signals** part, we compared our results with the software-based simulation results, which could be considered as the best scenario for CMOS-based results, as shown in **Fig. 4d** (re-drawn in **Figure R3** below). As we can see, only ~3% degradation in accuracy ($93.46\% \pm 0.44\%$ versus $96.41\% \pm 0.39\%$) has been observed in our memristor-based system, which has a much lower power consumption than typical CPU (e.g., ~65W TDP for Intel Core i7 8700) we used to run the software simulation. The small accuracy degradation can be attributed to the non-ideal device characteristics of the memristor array (e.g., fluctuation and noise in device conductance as shown in **Figure R4**), which could be alleviated by using a larger neural network (e.g., Li et al., *Nature Communications*, 2018) or new training strategies (e.g., Yao et al., *Nature*, 2020).

To clarify this point, we have added **Figure R4** as **Supplementary Figure 7** in the

revised **Supplementary Information** and also incorporated the above discussions in the revised manuscript:

On pages 8 - 9, lines 221 – 234: “**Similar to state-of-the-art CMOS designs, the effect of noise is considered in our memristor-based system. In the hardware implementation of our FIR filter bank and perceptron neural network using our memristor arrays, both the filter coefficients and synaptic weights are represented by the device conductance of memristors. Therefore, the noise effects, such as read noise, conductance variations and fluctuations, have already been incorporated in the filtered and classification results. For example, Supplementary Fig. 7 shows the memristor conductance distribution of eight typical conductance levels during mapping and read operations. In addition, the neural signals used in this work are adopted from the widely used Bonn Epilepsy Dataset, and they already captured background noises during real-world recording, including instrument noise and neural signal inherent noise. In the presence of those non-ideal factors, the achieved high accuracy in identifying epilepsy-related brain states in this work strongly indicates that our memristor-based system is robust to various noises in the hardware implementation.**”

Figure R3. Overall accuracies of the software-simulated and memristor array-implemented neural networks in the identification of the brain state. S.S. represents the network trained using software-calculated results. For M.S. and M.M., the networks are trained using memristor array-filtered results, and M.S. uses the software-simulated network while M.M. uses the memristor-implemented network for inference. e, Comparison of the power efficiency for CMOS-based and memristor-based systems.

Figure R4. Conductance distributions of memristors in the mapping and read operations. **a-h**, Mapped conductance distributions of eight typical levels: 2.5, 5.0, 7.5, 10.0, 12.5, 15.0, 17.5 and 20 μS . Each distribution includes the mapping results of 1000 memristors. **i-p**, Read noise distributions of these eight conductance levels. Each distribution is obtained from 100-times read results.

Comment #3

(3) The analog processing capability of the memristor-based electronics is used, but the Bonn Epilepsy dataset should be digital signals. How were these signals converted to analog in the first place? How were the 20 biomarkers/features extracted from the output currents of the filter banks? Could this extraction be implemented by the memristor-based electronics? If not, the concept of a memristor-based BMI is diminished. How were the input voltages representing the extracted biomarkers to the artificial neural network obtained from the features of the output currents of the filter banks?

Response:

We appreciate the concern raised by the reviewer. The analog processing capability of memristors refers the in-memory computations (i.e., multiplication and summing) that use analog values as inputs, rather than 0/1 representation in digital CMOS system. The neural signals in Bonn Epilepsy dataset (Andrzejak et al., *Phys Rev E Stat Nonlin Soft Matter Phys*, 2001) are sampled and digitalized in order to be stored and analyzed offline. In this demonstration, as **Figure R5** illustrates, the signals represented by the values with digitalized levels in the dataset are transformed to voltage pulses with analog amplitudes to be applied on the memristor array. The transformation is implemented by software and a custom system (Yao et al., *Nature*, 2020). To avoid any confusion, we have clarified this point in the revised manuscript (see **Methods**).

To extract the 20 biomarkers/features from the output currents of the filter banks, firstly, the output currents are converted by multiplication and offset to get the filtered voltage waveforms. Then we calculate the maximum value, minimum value, mean value, sum of absolute value and sum of energy for the voltage waveforms to obtain the 20 biomarkers (5 features for each of 4 waveforms). These computations are currently done by software in this work for simplicity but they could be potentially implemented by memristor-based electronics in the future. For example, there have already been attempts in literature to use memristors to carry out computations like calculating the maximum/minimum values (Amer et al., *2016 5th International Conference on Modern Circuits and Systems Technologies (MOCASST)*, 2016; Amer et al., *2015 IEEE International Conference on Electronics, Circuits, and Systems (ICECS)*, 2015), as illustrated in **Figure R6**. The calculations of mean value, sum of absolute value and sum of energy could be done similarly by converting them into multiplication and addition operations.

Furthermore, for the artificial neural network, we normalize the extracted biomarkers to the range of 0.1 ~ 0.3 V by a linear transformation (e.g., $y = a \times x + b$, x: input; y: output; a, b: amplification factor and offset) as the input voltages. The normalization for the biomarker vector is implemented by software in this work for simplicity. These calculations could also be implemented by either memristor or CMOS-based circuits (e.g., Li et al., *Nature Electronics*, 2018).

To clarify this point, we have added the above discussions in the revised manuscript to explain the extraction of biomarkers and the conversion of input voltages for the artificial neural network. We have added “In this demonstration, as **Supplementary Figure 8** illustrates, the signals represented by the values with digitalized levels in the dataset are transformed to voltage pulses with analog amplitudes to be applied on the memristor array.” on page 11, lines 273 - 276. Also, we have added **Figure R5** as **Supplementary Figure 8** in the revised **Supplementary Information**. We have added “to extract the 20 biomarkers/features from the output currents of the filter banks, firstly, the output currents are amplified and offset to obtain the filtered voltage waveforms. Then we calculate the maximum value, minimum value, mean value, sum of absolute value and sum of energy for the voltage waveforms to obtain the 20 biomarkers (5 features for each of 4 waveforms). These computations are currently done by software in this work for simplicity but in principle they could also be implemented by memristor-based electronics Amer et al., 2016 *5th International Conference on Modern Circuits and Systems Technologies (MOCASST)*, 2016; Amer et al., 2015 *IEEE International Conference on Electronics, Circuits, and Systems (ICECS)*, 2015. Furthermore, we normalize the extracted biomarkers to the range of 0.1 ~ 0.3 V by software using a linear transformation (e.g., $y = a \times x + b$, x : input; y : output; a , b : amplification factor and offset), and then use them as the input voltages for the memristor-based neural network.” on pages 12 – 13, lines 317 – 326.

Figure R5. Representation of neural signal inputs for memristor-based system. Sampled neural signals with digitalized values (a) in the Bonn Epilepsy Dataset are transformed to voltages pulses with analog amplitudes (b) and then applied on the memristor array. For example, a 121-point signal clip labeled as the interictal brain state is shown here. The pulse width is 50 ns in this work.

Figure R6. Memristor-based max-min circuits. a, the 2-input minimum circuits and its computation principle. Reversing the polarity of both memristors could implement

the maximum function. Adopted from Amer et al., *2016 5th International Conference on Modern Circuits and Systems Technologies (MOCAST)*, 2016 **b**, the 3-input minimum circuits. Adopted from Amer et al., *2015 IEEE International Conference on Electronics, Circuits, and Systems (ICECS)*, 2015.

Comment #4

(4) Equations 1-3 are for discrete-time, continuous-valued signals; particularly, Equation 1 is the discrete-time convolution sum. How were such discrete-time "analog" input signals ($x[n]$) obtained from the digital Bonn Epilepsy dataset.

Response:

Thank you for the comment. As we explained in the response to your last comment #3, the signals in Bonn Epilepsy dataset were transformed to discrete-time "analog" voltages pulses with identical duration (pulse width) and then applied to the memristor array through software and a custom system (Yao et al., *Nature*, 2020). In future fully integrated BMI system that use neural probes for signal recording, sample and hold circuits are required (but no analog-to-digital converters are needed) to directly obtain discrete-time "analog" neural signals as inputs for our memristor-based filter bank. This could simplify the signal transformation in the system implementation. To avoid any confusion, we have added **"In this demonstration, as Supplementary Figure 8 illustrates, the signals represented by the values with digitalized levels in the dataset are transformed to voltage pulses with analog amplitudes to be applied on the memristor array."** on page 11, lines 273 - 276 in the revised manuscript. Also, we have added **Figure R5 as Supplementary Figure 8** in the revised **Supplementary Information**.

Comment #5

(5) The first sentence of the Conclusion reads "In summary, we have proposed a novel BMI system...", however, such a BMI system implemented entirely or primarily using the memristor-based electronics is not demonstrated. Furthermore, no integration of signal-recording neural probes with signal-analysis memristor arrays is demonstrated.

Response:

Thanks for your comments. As we explained in the response to your first comment #1, **the main focus and novelty of this work is to experimentally demonstrate the feasibility of using memristor arrays to realize neural signal analysis and classification with high accuracy and power efficiency**, which is the most important component in our proposed BMI system. The reason why signal-recording neural probes are not experimentally integrated at this stage to implement the complete BMI system is as follows:

- 1) As we discussed in the Introduction part, **the performance and scalability of existing BMI systems is largely limited by the signal-processing electronics** rather than the signal-recording neural probes. Therefore, the goal of this work is to experimentally demonstrate the feasibility of using memristor arrays to implement both FIR filter bank and perceptron neural network, which realize neural signal analysis and classification with high accuracy and power efficiency.
- 2) To resemble real-world BMI applications, we use the neural signals from the widely used Bonn Epilepsy Dataset (e.g., Andrzejak et al., *Phys Rev E Stat Nonlin Soft*

Matter Phys, 2001; Adeli et al., *IEEE Transactions on Biomedical Engineering*, 2007; Orhan et al., *Expert Systems with Applications*, 2011; Samiee et al., *IEEE Transactions on Biomedical Engineering*, 2015; Acharya et al., *Computers in Biology and Medicine*, 2018), which are **raw neural signals recorded in real-world setting**, for the proof-of-concept demonstration for identifying epilepsy-related brain states using our memristor-based system.

- 3) Since algorithms like FIR filter are **generic** for various signal processing, **we expect our memristor-based system to be able to seamlessly work with many different types of neural probes**. In fact, inspired by the exciting results obtained in this work, we are currently working on integrating both neural probes and memristor arrays to build prototype BMI system for potential clinic trails, which is the very next step of the present work.

Therefore, we believe that, as the first and key step of implementing a complete BMI system, our work represents a significant milestone towards developing viable memristor-based modules for bio-inspired neural signal analysis. To make this point clear, we have revised the Conclusion part and incorporated the above discussions in the revised manuscript. We have incorporated the above discussions in the revised manuscript to further clarify the novelty and focus of this work as follows:

On page 1, line 21: “...a memristor-based neural signal analysis system, ...”; On page 1, line 23: “As a proof-of-concept demonstration, ...”; On page 3, lines 65 - 70: “we propose a memristor-based neural signal analysis system for next-generation BMIs. As a proof-of-concept demonstration, we use memristor arrays to implement both long-tap finite impulse response (FIR) filter bank (as a signal pre-processor) and perceptron neural network (as a decoder) in one model system to demonstrate filtering and identifying epilepsy-related brain activities. Owing to the excellent I-V linearity and analog switching behaviors of our memristors, the system ...”; On page 3, lines 76-78: “... the memristor array-based neural signal analysis system and the design of a complete BMI by integrating it with neural probes. ... in such a BMI as it ...”; On page 4, lines 86 - 89: “Since the preprocessor and decoder are typically the most critical and computation-extensive components in BMIs, their high-efficiency implementation would help enhance the performance and scalability of BMIs with multiple recoding sites.”; On pages 5 - 6, lines 113 – 124: “For the system demonstration, we use the neural signals from the widely used Bonn Epilepsy Dataset (see **Methods** for more information), which are LFP signals recorded in real-world setting, for identifying epilepsy-related brain states (normal, interictal and ictal) using our memristor-based system. The normal state indicates the subject is normal and has no epilepsy. Both interictal and ictal states could be observed from epilepsy patients. The former means the patient is during the interval between epileptic seizures while the latter shows an epileptic seizure is happening inside the brain. It should be noted that, neural probes, which are commercially available, are not experimentally integrated here to complete the BMI as they are not the focus of this study. Besides, since algorithms like FIR filter are generic for various signal processing, we expect our memristor-based system to be able to seamlessly work with many different types of neural probes.”; On page 10, lines

254–255: “..., we have proposed a memristor-based neural signal analysis system with high efficiency for future BMIs. Memristor ...”.

Reference:

- Acharya, U. R., S. L. Oh, Y. Hagiwara, J. H. Tan, and H. Adeli. 2018. 'Deep convolutional neural network for the automated detection and diagnosis of seizure using EEG signals', *Computers in Biology and Medicine*, 100: 270-78.
- Adeli, H., S. Ghosh-Dastidar, and N. Dadmehr. 2007. 'A wavelet-chaos methodology for analysis of EEGs and EEG subbands to detect seizure and epilepsy', *IEEE Transactions on Biomedical Engineering*, 54: 205-11.
- Alibart, F., L. Gao, and D. Strukov. 2016. 'A Reconfigurable FIR Filter with Memristor-Based Weights', *arxiv.1608.05445*.
- Ambrogio, Stefano, Pritish Narayanan, Hsin-yu Tsai, Robert M. Shelby, Irem Boybat, Carmelo di Nolfo, Severin Sidler, Massimo Giordano, Martina Bodini, Nathan C. P. Farinha, Benjamin Killeen, Christina Cheng, Yassine Jaoudi, and Geoffrey W. Burr. 2018. 'Equivalent-accuracy accelerated neural-network training using analogue memory', *Nature*, 558: 60-67.
- Amer, S. H., A. H. Madian, H. ElSayed, and A. S. Emara. 2016. "Effect of the memristor threshold current on memristor-based Min-Max circuits." In *2016 5th International Conference on Modern Circuits and Systems Technologies (MOCASST)*, 1-4.
- Amer, S. H., A. H. Madian, and A. S. Emara. 2015. "Design and analysis of memristor-based min-max circuit." In *2015 IEEE International Conference on Electronics, Circuits, and Systems (ICECS)*, 187-90.
- Andrzejak, R. G., K. Lehnertz, F. Mormann, C. Rieke, P. David, and C. E. Elger. 2001. 'Indications of nonlinear deterministic and finite-dimensional structures in time series of brain electrical activity: dependence on recording region and brain state', *Phys Rev E Stat Nonlin Soft Matter Phys*, 64: 061907.
- Bajaj, V., and R. B. Pachori. 2012. 'Classification of Seizure and Nonseizure EEG Signals Using Empirical Mode Decomposition', *Ieee Transactions on Information Technology in Biomedicine*, 16: 1135-42.
- Crispin-Bailey, C., C. L. Dai, and J. Austin. 2019. 'A 65-nm CMOS Lossless Bio-Signal Compression Circuit With 250 FemtoJoule Performance Per Bit', *IEEE Transactions on Biomedical Circuits and Systems*, 13: 1087-100.
- Daoud, Hisham, and Magdy Bayoumi. 2020. 'Deep Learning Approach for Epileptic Focus Localization', *IEEE Transactions on Biomedical Circuits and Systems*, 14: 209-20.
- Guler, I., and E. D. Ubeyli. 2005. 'Adaptive neuro-fuzzy inference system for classification of EEG signals using wavelet coefficients', *Journal of Neuroscience Methods*, 148: 113-21.
- Li, Can, Daniel Belkin, Yunning Li, Peng Yan, Miao Hu, Ning Ge, Hao Jiang, Eric Montgomery, Peng Lin, Zhongrui Wang, Wenhao Song, John Paul Strachan, Mark Barnell, Qing Wu, R. Stanley Williams, J. Joshua Yang, and Qiangfei Xia. 2018. 'Efficient and self-adaptive in-situ learning in multilayer memristor neural networks', *Nature Communications*, 9: 2385.
- Li, Can, Miao Hu, Yunning Li, Hao Jiang, Ning Ge, Eric Montgomery, Jiaming Zhang, Wenhao Song, Noraica Dávila, Catherine E. Graves, Zhiyong Li, John Paul Strachan, Peng Lin, Zhongrui Wang, Mark Barnell, Qing Wu, R. Stanley Williams, J. Joshua Yang, and Qiangfei Xia. 2018. 'Analogue signal and image processing with large memristor crossbars', *Nature Electronics*, 1: 52-59.
- Li, Can, Zhongrui Wang, Mingyi Rao, Daniel Belkin, Wenhao Song, Hao Jiang, Peng Yan, Yunning Li, Peng Lin, Miao Hu, Ning Ge, John Paul Strachan, Mark Barnell, Qing Wu, R. Stanley Williams,

- J. Joshua Yang, and Qiangfei Xia. 2019. 'Long short-term memory networks in memristor crossbar arrays', *Nature Machine Intelligence*, 1: 49-57.
- Mirebrahimi, Seyedeh-Nafiseh, and Farshad Merrikh-Bayat. 2014. 'Programmable discrete-time type I and type II FIR filter design on the memristor crossbar structure', *Analog Integr Circ Sig Process*, 79: 529-41.
- Nourazar, M., V. Rashtchi, F. Merrikh-Bayat, and A. Azarpeyvand. 2018. 'Towards memristor-based approximate accelerator: application to complex-valued FIR filter bank', *Analog Integrated Circuits and Signal Processing*, 96: 577-88.
- Orhan, U., M. Hekim, and M. Ozer. 2011. 'EEG signals classification using the K-means clustering and a multilayer perceptron neural network model', *Expert Systems with Applications*, 38: 13475-81.
- Samiee, K., P. Kovacs, and M. Gabbouj. 2015. 'Epileptic Seizure Classification of EEG Time-Series Using Rational Discrete Short-Time Fourier Transform', *IEEE Transactions on Biomedical Engineering*, 62: 541-52.
- Sun, Zhong, Giacomo Pedretti, Elia Ambrosi, Alessandro Bricalli, Wei Wang, and Daniele Ielmini. 2019. 'Solving matrix equations in one step with cross-point resistive arrays', *Proceedings of the National Academy of Sciences*, 116: 4123.
- Wang, Zhongrui, Can Li, Wenhao Song, Mingyi Rao, Daniel Belkin, Yunning Li, Peng Yan, Hao Jiang, Peng Lin, Miao Hu, John Paul Strachan, Ning Ge, Mark Barnell, Qing Wu, Andrew G. Barto, Qinru Qiu, R. Stanley Williams, Qiangfei Xia, and J. Joshua Yang. 2019. 'Reinforcement learning with analogue memristor arrays', *Nature Electronics*, 2: 115-24.
- Yao, Peng, Huaqiang Wu, Bin Gao, Jianshi Tang, Qingtian Zhang, Wenqiang Zhang, J. Joshua Yang, and He Qian. 2020. 'Fully hardware-implemented memristor convolutional neural network', *Nature*, 577: 641-46.

REVIEWERS' COMMENTS:

Reviewer #1 (Remarks to the Author):

Thank you for considering the comments.

Reviewer #2 (Remarks to the Author):

The revisions are largely satisfactory. But what needs to be emphasized is that the claims on contribution and novelty need to be supported by the actual data reported. Some extending discussions on implications of the data/results are good additions, but overstatements should be avoided, particularly in the Abstract, Introduction and Conclusion.

Response Letter to Reviewers' Comments

Reviewer #1 (Remarks to the Author):

Thank you for considering the comments.

Response:

Thank you very much for the valuable time you have spent reviewing our manuscript and providing insightful comments to help significantly improve the quality of our work. We are very glad to see that you are satisfied with our revision.

Reviewer #2 (Remarks to the Author):

The revisions are largely satisfactory. But what needs to be emphasized is that the claims on contribution and novelty need to be supported by the actual data reported. Some extending discussions on implications of the data/results are good additions, but overstatements should be avoided, particularly in the Abstract, Introduction and Conclusion.

Response:

Thank you very much for the valuable time you have spent reviewing our manuscript and providing insightful comments to help significantly improve the quality of our work. We are glad to hear that you are largely satisfied with our revision. As to your one last comment, we have carefully revised any potential overstatements in the manuscript regarding the novelty and contribution of this work. Major revisions are summarized as follows:

- 1) We have revised the **Abstract** to avoid any overstatements. The last sentence now reads: *This work demonstrates the feasibility of using memristors for high-performance neural signal analysis in next-generation brain-machine interfaces.*
- 2) We have also revised the **Conclusion (Discussion)** section, and changed the “Our work experimentally demonstrates the tremendous potential of...” to “Our work experimentally demonstrates the potential of...”.